# Successful Implementation of HITOC and HIPEC in the Management of Advanced Ovarian Carcinoma with Pleural and Peritoneal Carcinomatosis

**DOI:** 10.3390/diagnostics14050455

**Published:** 2024-02-20

**Authors:** Bogdan Moldovan, Codin Theodor Saon, Iris-Iuliana Adam, Radu-Mihai Pisica, Vlad Teodor Silaghi, Vlad Untaru, Doly Stoica, Madalina Crisan, Andreea Popianas, Florentina Pescaru, Adriana Zolog, Liliana Vecerzan

**Affiliations:** 1‘St. Constantin’ Hospital, 500299 Brasov, Romania; codints@gmail.com (C.T.S.); radu.pisica@spitalulsfconstantin.ro (R.-M.P.); silaghivladteodor@gmail.com (V.T.S.); vlad.untaru@spitalulsfconstantin.ro (V.U.); stoicadoly@yahoo.com (D.S.); madalina.crisan@spitalulsfconstantin.ro (M.C.); dr.popianas@gmail.com (A.P.); florentina.pescaru@spitalulsfconstantin.ro (F.P.); 2Faculty of Medicine, “Lucian Blaga” University, 550169 Sibiu, Romania; adamirisiuliana@gmail.com (I.-I.A.); liliana.novac@ulbsibiu.ro (L.V.); 3Pathology Department, Regina Maria Hospital, 400500 Cluj-Napoca, Romania; adizolog@yahoo.com

**Keywords:** HITOC, HIPEC, hyperthermia, cytoreductive surgery, carcinomatosis, ovarian cancer, cisplatin, doxorubicin

## Abstract

This case report details the application and outcomes of a novel therapeutic approach involving hyperthermic intraperitoneal chemotherapy (HIPEC) and hyperthermic intrathoracic chemotherapy (HITOC) in a single patient diagnosed with advanced ovarian neoplasm. The treatment protocol included pleural cytoreductive surgery (CRS) and HITOC followed by a second surgical intervention consisting of peritoneal CRS and HIPEC. HIPEC targeted the intraperitoneal space with heated chemotherapy, while HITOC extended the thermal perfusion to the thoracic cavity. The patient has shown significant progression in disease-free survival over one year and eight months of observation, demonstrating lower recurrence rates and an overall survival outcome exceeding expectations based on conventional therapy outcomes. The combined modality demonstrated a manageable toxicity profile, with no significant increase in peri- or postoperative complications observed.

The development of therapeutic strategies for advanced abdominal and thoracic malignancies has progressed significantly, leading to the emergence of innovative modalities like hyperthermic intraperitoneal chemotherapy (HIPEC) and hyperthermic intrathoracic chemotherapy (HITOC). Originating in the mid-20th century, these approaches were pioneered to tackle the challenges posed by peritoneal and thoracic malignancies resistant to conventional treatments. Experimental work in the late 1940s laid the foundation for hyperthermic chemotherapy, utilizing elevated temperatures to enhance drug penetration, cytotoxicity, and apoptotic responses within malignant tissues [1,2]. The application of HIPEC gained prominence in the 1980s, particularly for diseases like pseudomyxoma peritonei and peritoneal mesothelioma. This success prompted the extension of hyperthermic techniques to intrathoracic malignancies with the advent of HITOC, specifically addressing challenges in ovarian-related thoracic diseases [1,3]. HITOC adapts the principles of hyperthermia and localized chemotherapy delivery to optimize the treatment of intrathoracic diseases through targeted thermal chemoperfusion.

A 58-year-old female with a medical history of a non-resectable, high-grade serous ovarian carcinoma, initially diagnosed and treated with eight cycles of carboplatin and paclitaxel in 2019, presented with a recurrence of symptoms, including abdominal discomfort, distension, and dyspnea. The initial biopsy performed in 2019 confirmed the presence of positive breast cancer gene (BRCA), and homologus recombination deficiency (HRD) mutation status was wild type, indicating that this mutation was identified at the time of the initial diagnosis and justifying the initial course of treatment. Considering the patient’s history of non-resectable, high-grade serous ovarian carcinoma and positive BRCA mutation status, the maintenance treatment option chosen for this case after the initial eight cycles of carboplatin and paclitaxel involved PARP inhibitors, given their efficacy in targeting BRCA-mutated ovarian cancer. The clinical and paraclinical examination from 2022 revealed the presence of ascites and pleuritic chest pain, prompting further investigation into the possibility of disease recurrence.

Physical examination disclosed a distended abdomen with fluid wave, indicative of ascites. The patient reported recent-onset dyspnea and was noted to have reduced breath sounds on auscultation, raising suspicion of pleural involvement. Imaging studies, including thoracic, abdominal, and pelvic computed tomography (CT) scans, revealed extensive peritoneal carcinomatosis (Figure 1 and Figure 2) with notable ascites and pleural effusion (Figure 3).

Subsequently, after proper surgical evaluation, it was decided that a combined surgical therapy was needed, consisting of pleurectomy and the decortication of the right pleura (Figure 4) and HITOC lavage (Figure 5) at 42 degrees Celsius for 60 min using cisplatin 150 mg/m^2^ and doxorubicin 20 mg/m^2^ for the first surgical intervention. The postoperative course following pleural CRS and HITOC was remarkably favorable, marked by improved respiratory function, controlled pain, and symptomatic relief allowing the patient’s discharge 4 days postoperatively. The histopathological report revealed pleural metastases with features of high-grade serous carcinoma (G2) within the examined material, which, in the clinical context, may have ovarian origin (Figure 6).

The absence of complications contributed to the overall positive trajectory of the patient’s recovery, allowing the second surgical intervention to be performed, which consisted of peritoneal CRS and HIPEC one month later. It consisted of an en bloc posterior pelvic exenteration with a pelvic peritonectomy [1,2,3,4,5,6,7,8,9,10], lymphadenectomy, and en bloc enlarged enterectomy with a total colectomy and terminal ileostomy (Figure 7 and Figure 8). It is worth noting that lymphadenectomy plays a crucial role in peritoneal carcinomatosis cases by providing essential insights into regional lymph node involvement in advanced disease forms, aiding in accurate staging, guiding therapeutic decisions, and enhancing prognostic assessments. The HIPEC lavage was performed using cisplatin 100 mg/m^2^ and doxorubicin 50 mg/m^2^ at 42 degrees Celsius for 90 min [6,7,8,9,10,11]. The patient’s postoperative evolution after HIPEC was marked by a combination of physiological stability, effective pain management, restored organ functions, and an overall improvement in wellbeing. Regular follow-up and ongoing surveillance were crucial for a long-term positive outcome. The findings outlined in the second histopathological report align with metastases in the peritoneum, abdominal wall, intestines, and lymph nodes, indicating high-grade serous carcinoma (G2), which, clinically, is suggestive of ovarian origin (Figure 6). The patient was tested for BRCA and HRD mutations, which suggested a wild type. The proposed chemotherapeutical treatment that was agreed on was the use of carboplatin and paclitaxel. A platinum-free interval (TFIp) exceeding one year was observed between the initial treatment and the diagnosis of recurrence, indicating a substantial period without platinum exposure, allowing the patient to undergo a new treatment with platinum-based chemotherapy.

A landmark achievement in the treatment of pleural and peritoneal carcinomatosis for ovarian cancers was reached by our team in 2022, involving a unique combination of hyperthermic intrathoracic chemotherapy (HITOC) and hyperthermic intraperitoneal chemotherapy (HIPEC), strategically performed one month apart. The decision to conduct peritoneal surgery a month after thoracic surgery was influenced by the patient’s presentation of complete right lung collapse and respiratory failure, prompting the initial choice of hyperthermic intrathoracic chemotherapy (HITHOC). The one-month interval was arbitrary and necessary for the patient’s recovery. Our multidisciplinary team who was behind this innovative strategy aims to optimize cytoreduction and enhance the effectiveness of chemotherapy.

HITOC and HIPEC address both thoracic and peritoneal manifestations of ovarian cancer [12], providing comprehensive coverage that targets potential tumor deposits in multiple anatomical regions. Our hypothesis suggests that the induction of hyperthermic conditions during both procedures could potentially enhance cytoreduction, facilitating the more efficient removal of visible tumor deposits. This can contribute to improved surgical outcomes and potentially increase the success of subsequent chemotherapy. By administering heated chemotherapy directly to the affected areas (thoracic and peritoneal cavities), HITOC and HIPEC maximize the concentration and efficacy of the chemotherapy agents while minimizing systemic exposure. This localized approach helps spare healthy tissues from unnecessary toxicity [12].

The sequential application of HITOC and HIPEC may have a synergistic effect [11], as the heated chemotherapy targets microscopic residual disease that may be present after cytoreduction. This combination aims to enhance the overall antitumor impact. After a six-week recovery period following the HIPEC procedure, the patient was referred to the oncology department for adjuvant chemotherapeutic treatment. As of one year and eight months after HITOC and HIPEC, the patient’s survival rate is indicative of a positive response to the implemented treatment regimen, emphasizing encouraging prospects for long-term wellbeing [13,14]. In our service, a significant milestone was reached, with a 40% 5-year survival rate for ovarian peritoneal carcinomatosis. The ongoing efforts in managing cases are highlighted by the current patient, who has had a 1.8-year survival without signs of recurrence, reflecting our commitment to advancing outcomes in ovarian peritoneal carcinomatosis.

The patient showed significant improvements in disease-free survival (currently 1 year and 8 months), reduced recurrence rates, and significant improvements in overall survival outcomes compared to outcomes expected with conventional treatment. No significant increases in intraoperative or postoperative complications were observed with this combination therapy, demonstrating a manageable toxicity profile [15,16,17,18,19].

In conclusion, the case report highlights the successful implementation of a comprehensive treatment strategy involving CRS-HITOC followed by CRS-HIPEC in a 58-year-old patient with ovarian cancer and peritoneal and pleural carcinomatosis. The staged approach, with HITOC as the first surgery followed by HIPEC a month later, demonstrated favorable postoperative evolution [20,21] (Figure 9). The use of these hyperthermic chemoperfusion techniques exhibited promising results in controlling disease progression and managing symptoms associated with advanced ovarian cancer. This pioneering case not only underscores the potential efficacy of HITOC and HIPEC in addressing multifocal disease but also emphasizes the importance of a multidisciplinary approach in the evolving landscape of ovarian cancer management. Continued research and the accumulation of similar case experiences will contribute to further defining the role of HITOC and HIPEC in the integrated treatment of advanced ovarian malignancies [18,19,20,21].

## Figures and Tables

**Figure 1 diagnostics-14-00455-f001:**
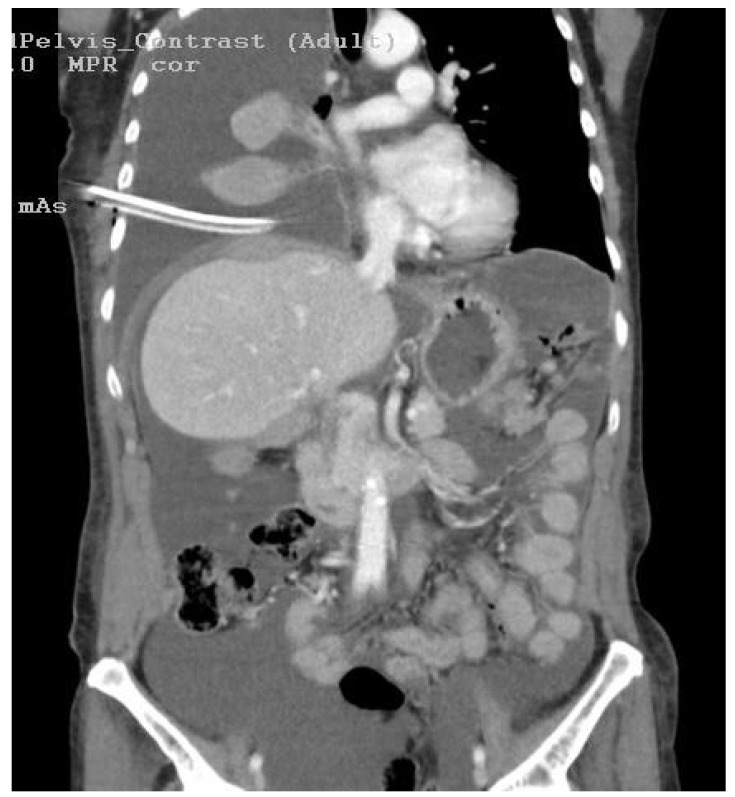
This coronal computed tomography (CT) image provides a detailed clinical insight into a complex medical scenario. Notably, it reveals a conspicuous right-sided pleural effusion (A), characterized by a substantial accumulation of fluid within the pleural cavity. Simultaneously, the image captures the presence of peritoneal carcinomatosis, suggesting the involvement of malignant cells within the peritoneum. The intricate interplay of these two findings signifies a complex disease state, likely reflective of an advanced stage of malignancy. The peritoneal carcinomatosis raises the suspicion of metastatic spread.

**Figure 2 diagnostics-14-00455-f002:**
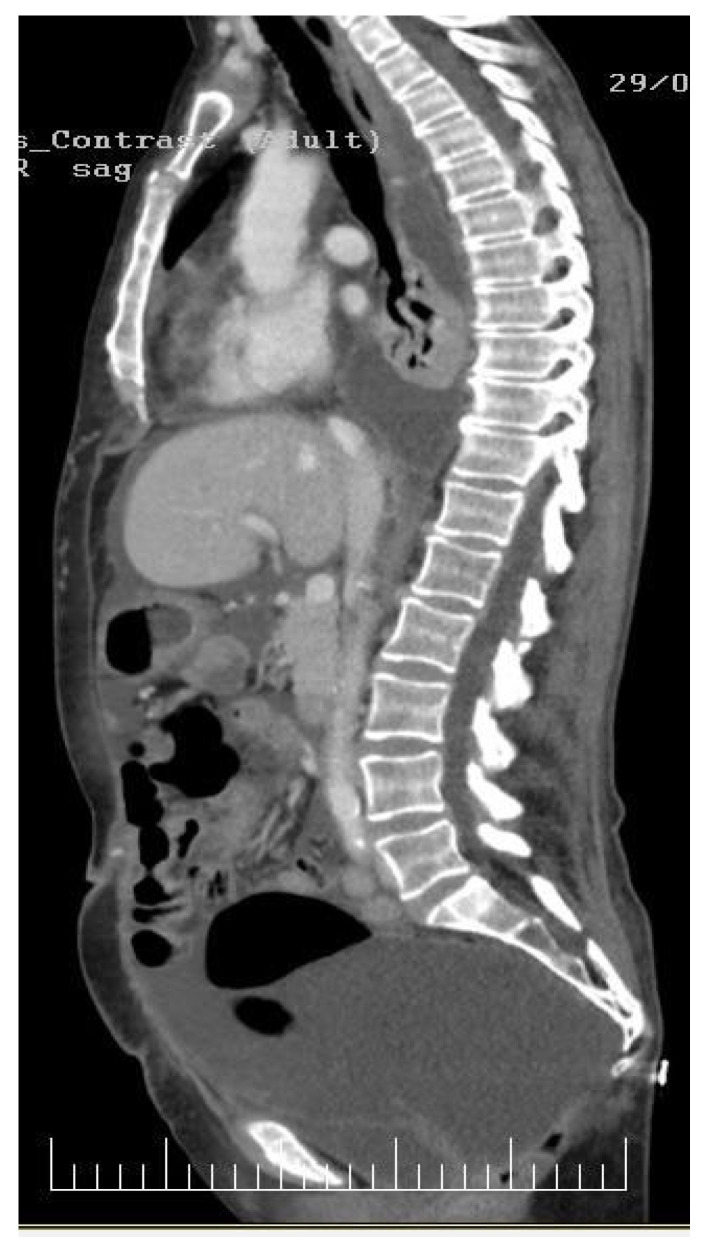
In this sagittal CT image, a comprehensive depiction emerges, revealing diffused involvement of both the pleura and peritoneum by carcinomatosis. The imaging highlights subtle yet pervasive changes in the pleural and peritoneal spaces, indicating the presence of malignant cells. Within the pleural cavity, there is evidence of diffused pleural thickening and irregularities. These findings suggest the infiltration of cancer cells along the pleural surfaces, impacting the normal architecture of the pleura. The diffuse nature of these changes implies an advanced stage of carcinomatosis within the thoracic compartment. Simultaneously, the peritoneal cavity exhibits similar diffused changes, with discernible thickening and irregularities along the peritoneal surfaces. This implies widespread involvement of the peritoneum by carcinomatosis.

**Figure 3 diagnostics-14-00455-f003:**
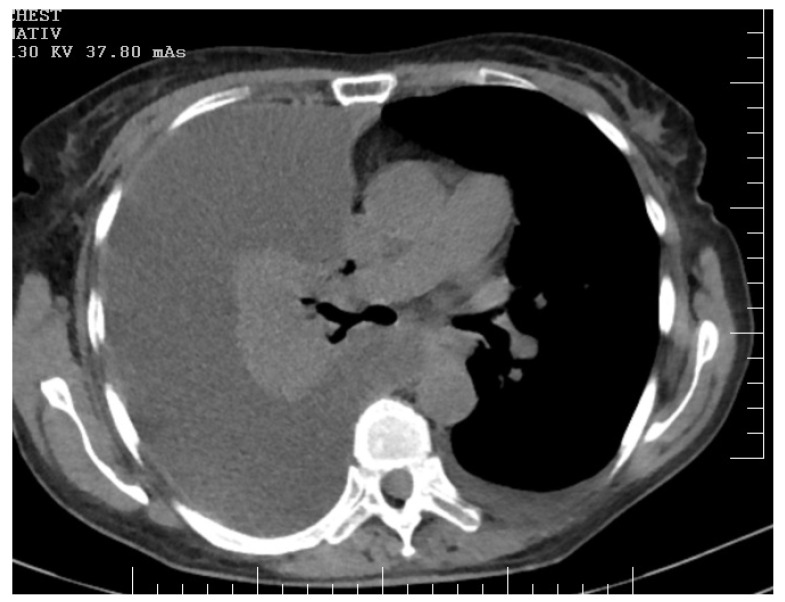
CT offers a detailed view of a clinically significant condition marked by a massive right-sided pleural effusion and accompanying total collapse of the right lung. The density and homogeneity of the effusion suggest a significant volume, resulting in marked compression and displacement of adjacent thoracic structures. The mediastinal structures are visibly shifted towards the left hemithorax, indicative of the expansive nature of the pleural effusion. Concurrently, the collapsed right lung is evident as a hyperdense mass with a noticeable reduction in volume, signifying atelectasis encompassing the upper, middle, and lower lobes.

**Figure 4 diagnostics-14-00455-f004:**
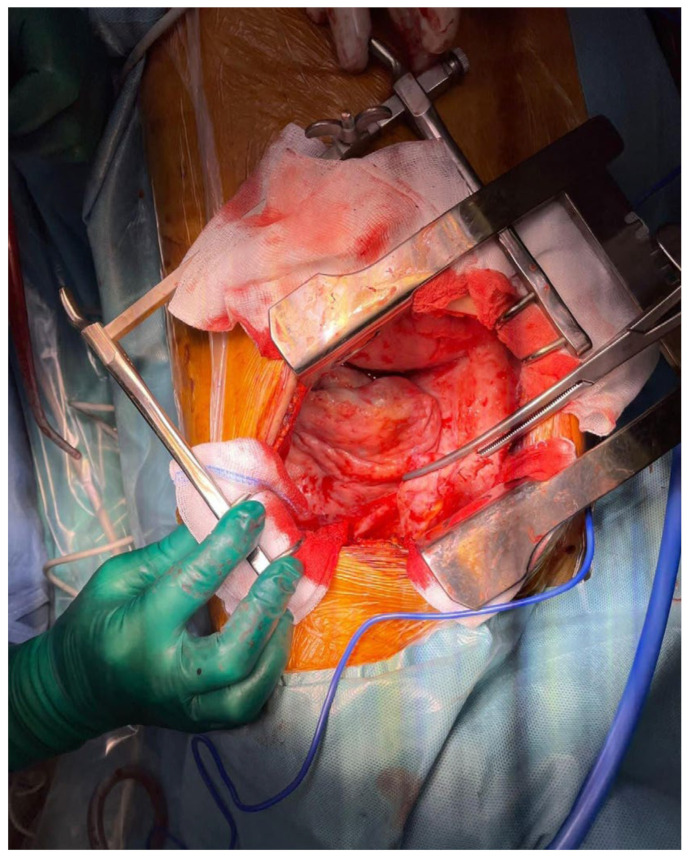
Pleural decortication final aspect. Post-decortication, the pleura appeared smoother and more pliable, indicating the successful removal of fibrous or inflammatory tissue that had encased the lung. The lung itself exhibited improved expansion, with the re-establishment of normal respiratory movement.

**Figure 5 diagnostics-14-00455-f005:**
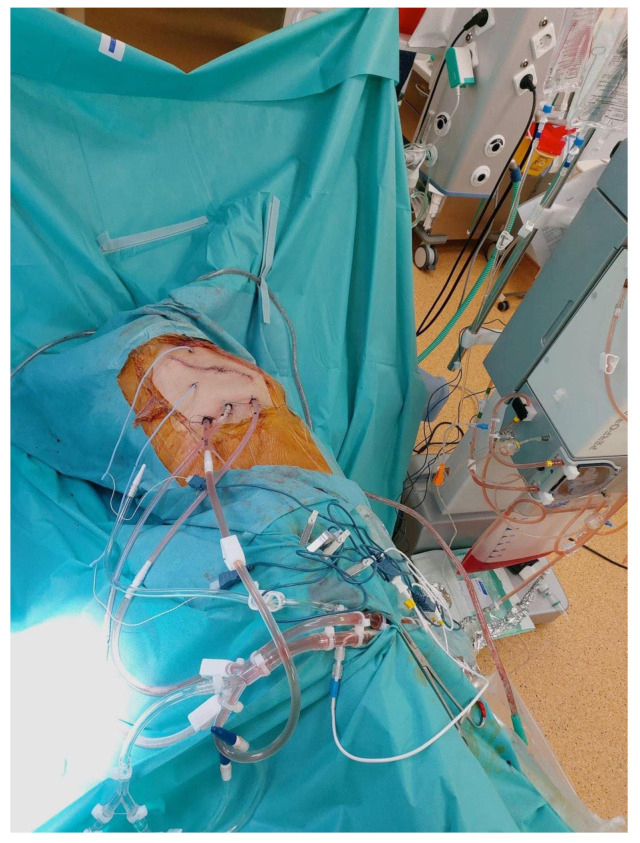
HITOC lavage at 42 degrees Celsius for 60 min using cisplatin 150 mg/m^2^ and doxorubicin 20 mg/m^2^.

**Figure 6 diagnostics-14-00455-f006:**
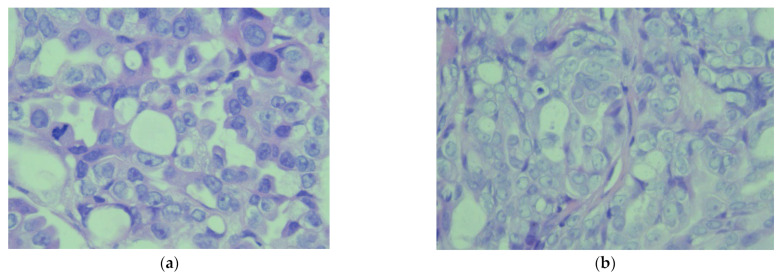
(**a**) Microscopic image (×40) from the histopathological examination showing the peritoneal tumor. The tumor cells are medium-sized, polygonal, without clear cellular boundaries, with moderately eosinophilic cytoplasm, and with a euchromatic nucleus. Some have a prominent nucleolus, focal accentuation of nuclear atypia, and frequent mitoses, both typical and atypical. (**b**) Microscopic image (×40) from the histopathological examination which shows the pleural tumor. The tumor cells are medium-sized and polygonal, without clear cellular boundaries, with moderately eosinophilic cytoplasms in moderate quantity, and with a euchromatic nuclei, of which some present a prominent nucleolus.

**Figure 7 diagnostics-14-00455-f007:**
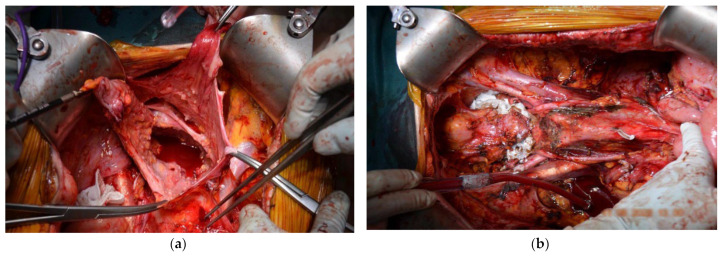
(**a**) Initial intraoperative aspect showing signs of diffused peritoneal carcinomatosis showing thickening and irregularities along the peritoneal surfaces. (**b**) Final operative look before suture and HIPEC lavage.

**Figure 8 diagnostics-14-00455-f008:**
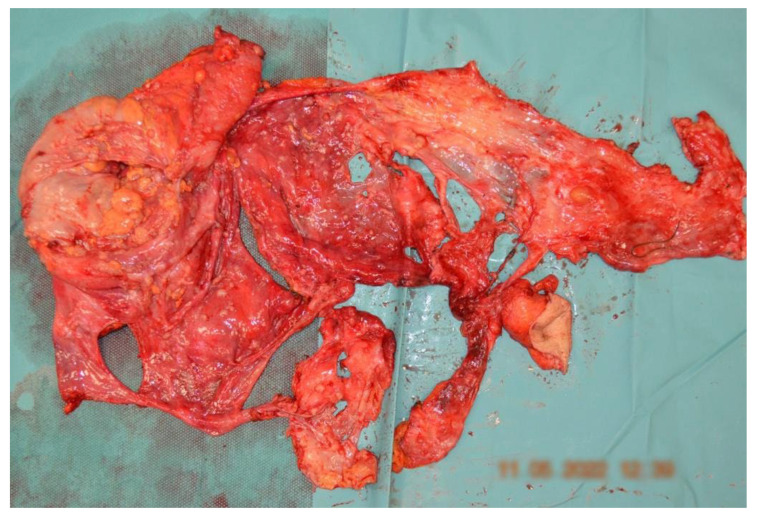
Surgical specimen from the peritoneal CRS presenting en bloc posterior pelvic exenteration with a pelvic peritonectomy.

**Figure 9 diagnostics-14-00455-f009:**
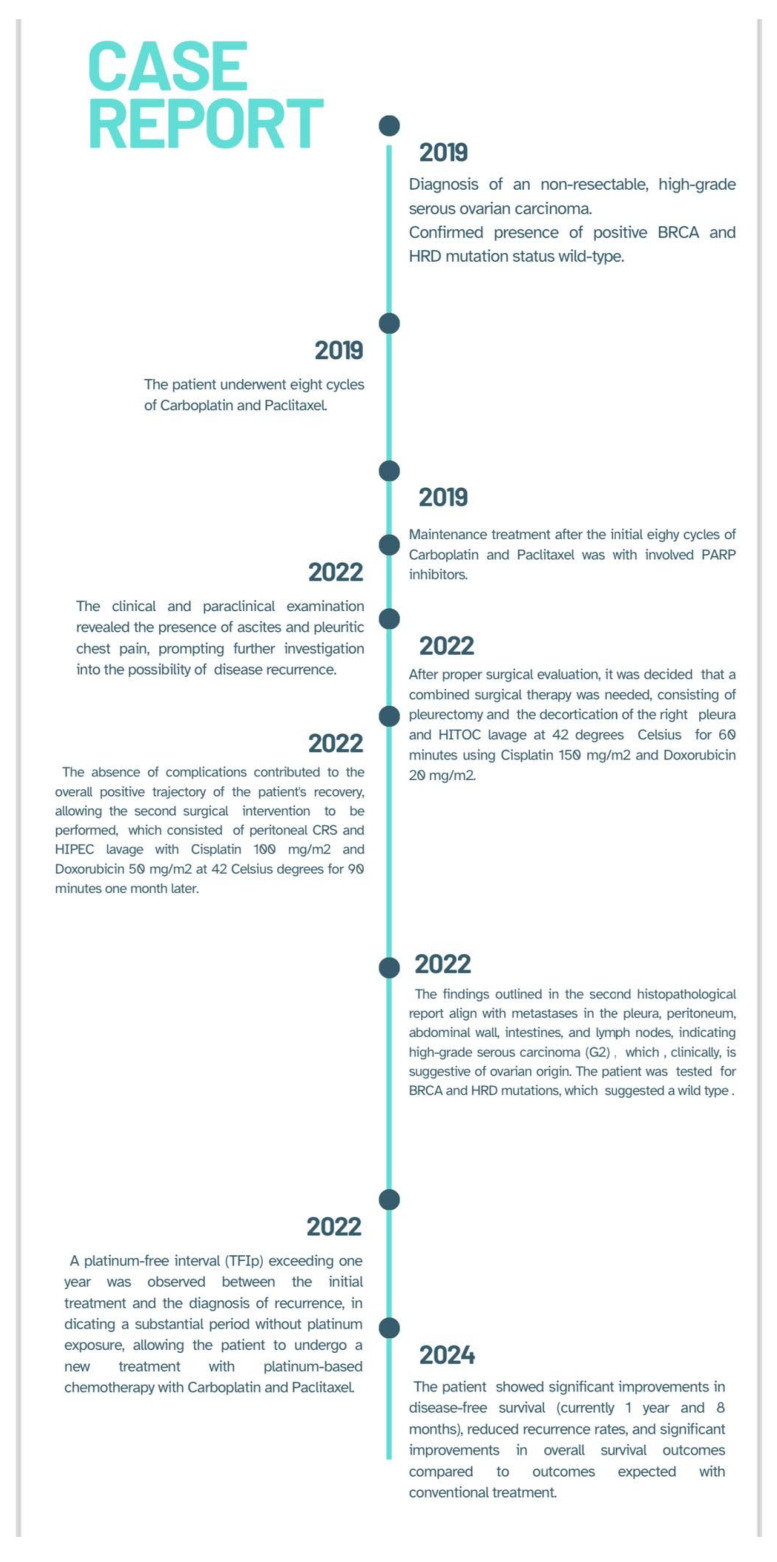
Case report timeline containing a chronological summary of the patient’s medical history regarding the oncological diagnosis, treatment, and surgical procedures.

## Data Availability

Not applicable.

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
