# Peer review of "Successful Implementation of HITOC and HIPEC in the Management of Advanced Ovarian Carcinoma with Pleural and Peritoneal Carcinomatosis"

_diagnostics, 2024, doi:10.3390/diagnostics14050455_

Round 1

Reviewer 1 Report

Comments and Suggestions for Authors

First, I would like to thank the reviewers on reporting this Advanced ovarian cancer case. I have some remarks for the authors after revising the entire manuscript:

-          Abstract

I would suggest the authors to rephrase the following sentence: “The patient exhibited a notable improvement in disease-free survival currently at one year and eight months, a reduction in recurrence rates, and an overall survival outcome compared to anticipated outcomes with conventional therapy”. As there is no comparation between groups reported (it is a case report), stating that “a reduction in recurrence or overall survival” has been observed seems too straightforward for the level of the evidence obtained.

-          Manuscript:

-          Why was the patient treated off-protocol with 8 cycles of Carboplatin-Paclitaxel, without any surgical attempt? Was the patient considered non resectable upfront?

-          Was the mutation status (HRD status, BRCA) tested before or after secondary cytoreductive surgery? If it was tested after, why wasn’t it performed during 1st course of treatment?

-          Which was the histology considered in the first place?

-          How much Platinum free interval (TFIp) was observed between 1st treatment and diagnosis of the recurrence?

-          Why was the peritoneal surgery performed 1 month after the thoracic surgery? What extension did the tumour have at second recurrence? Is lymphadenectomy necessary in this context?

-          . The hyperthermic conditions during both procedures aid in cytoreduction, promoting more effective removal of visible tumor deposits.” This is a hypothesis, not a fact. Please rephrase.

-          “This can contribute to improved surgical outcomes and potentially increase the success of subsequent chemotherapy”. On what grounds do the autors manifest this sentence? Is there any scientific analysis prior to this report?

-          “The sequential application of HITOC and HIPEC may have a synergistic effect [10], as the heated chemotherapy targets microscopic residual disease that may be present after cytoreduction. This combination aims to enhance the overall antitumor impact.” I cannot understand why sequential procedures may bear more prognostic benefit than concomitant ones.

-          “Focusing chemotherapy delivery to specific anatomical areas reduces systemic exposure to the drugs, potentially minimizing side effects commonly associated with traditional intravenous chemotherapy”. This sentence is repeated from the previous paragraph.

-          To publish personal images in a case report, authors must have prior consent from the patient. Was it obtained?

Overall comment on the Manuscript: From my point of view, the report lacks key data of the clinical case, mainly TFIp, reason for not performing surgery in the first place, reason for delivering 8 cycles of Carboplatin-Paclitaxel, maintenance treatment, and so on (tumoral markers at diagnosis, at recurrence diagnosis, etc). These pieces of information are key to understand or justify why a second cytoreductive surgery was proposed.

Nowadays, for a patient not included in a clinical trial, there is no justification to a) receive a second cytoreductive surgery aiming at the pleural and abdominal cavity and b) to receive HIPEC (nor HITOC), as recent clinical trials have failed to demonstrate a benefit in this context (Zivanovic, 2021), being only acceptable if performed in a “postCT” setting (CHIPOR Trial, Classe 2023). The authors did not justify correctly the rationale behind delaying 1mo the abdominal surgery after the thoracic surgery.

Comments on the Quality of English Language

Moderate English editing is needed. 

Author Response

Comment 1: 

Abstract

I would suggest the authors to rephrase the following sentence: “The patient exhibited a notable improvement in disease-free survival currently at one year and eight months, a reduction in recurrence rates, and an overall survival outcome compared to anticipated outcomes with conventional therapy”. As there is no comparation between groups reported (it is a case report), stating that “a reduction in recurrence or overall survival” has been observed seems too straightforward for the level of the evidence obtained.

Response 1: Thank you for pointing this out. I/We agree with this comment. Therefore, we have rephrased the sentence.

Comment 2: Why was the patient treated off-protocol with 8 cycles of Carboplatin-Paclitaxel, without any surgical attempt? Was the patient considered non resectable upfront?

Response 2: We have, accordingly, revised this matter. The patient recieved 8 cycles of Carboplatin-Paclitaxel because of the positive BRCA mutation and because of the fact that the tumor was considered non-resectable at the diagnosis time. The BRCA and HRD mutations were tested from the first biopsy performed to the patient where the results came back confirming a high-grade ovarian serous carcinoma. We detailed this matter when describing the case report.

Comment 3: Was the mutation status (HRD status, BRCA) tested before or after secondary cytoreductive surgery? If it was tested after, why wasn’t it performed during 1st course of treatment?

Response 3: We have, accordingly, revised this matter. The BRCA and HRD mutations were tested from the first biopsy performed to the patient where the results came back confirming a high-grade serous ovarian carcinoma. The BRCA mutation was positive and the HRD mutation status was a wild-type.

Comment 4: Which was the histology considered in the first place?

Response 4: We have, accordingly, revised this matter. The initial histology presented a high-grade serous ovarian carcinoma.

Comment 5: How much Platinum free interval (TFIp) was observed between 1st treatment and diagnosis of the recurrence?

Response 5: We have, accordingly, revised this matter. The TFIp interval exceeding one year was observed between the initial treatment and the diagnosis of recurrence.

Comment 6: Why was the peritoneal surgery performed 1 month after the thoracic surgery? What extension did the tumour have at second recurrence? Is lymphadenectomy necessary in this context?

Response 6: We have, accordingly, revised this matter. 

Comment 7:  “. The hyperthermic conditions during both procedures aid in cytoreduction, promoting more effective removal of visible tumor deposits.” This is a hypothesis, not a fact. Please rephrase.

Response 7: We have, accordingly, revised this matter. The decision to conduct peritoneal surgery a month after thoracic surgery was influenced by the patient’s presentation of complete right lung collapse and respiratory failure, prompting the initial choice of hyperthermic intrathoracic chemotherapy (HITHOC). The one-month interval was arbitrary and necessary for the patient’s recovery.

Comment 8:  “This can contribute to improved surgical outcomes and potentially increase the success of subsequent chemotherapy”. On what grounds do the autors manifest this sentence? Is there any scientific analysis prior to this report?

Response 8: We have, accordingly, revised this matter. If the debulking procedure is done correctly both thoracic and peritoneal, and the HIHOC/HIPEC lavage acts on the possible remaining microscopic structures, chemotherapy will be more efficient because the carcinomatosis will not interfere anymore with the treatment.

Comment 9: “The sequential application of HITOC and HIPEC may have a synergistic effect [10], as the heated chemotherapy targets microscopic residual disease that may be present after cytoreduction. This combination aims to enhance the overall antitumor impact.” I cannot understand why sequential procedures may bear more prognostic benefit than concomitant ones.

Response 9: We have, accordingly, revised this matter. For our case this was the best strategy beacause of the respiratory problems she presented; complete right lung collapse and respiratory failure, prompting the initial choice of hyperthermic intrathoracic chemotherapy (HITHOC). The one-month interval was arbitrary and necessary for the patient’s recovery.

Comment 10:  “Focusing chemotherapy delivery to specific anatomical areas reduces systemic exposure to the drugs, potentially minimizing side effects commonly associated with traditional intravenous chemotherapy”. This sentence is repeated from the previous paragraph.

Response 10: We have, accordingly, revised this matter.

Comment 11: To publish personal images in a case report, authors must have prior consent from the patient. Was it obtained?

Response 11: We have, accordingly, revised this matter. The consent form was uploaded from the begining when we submited the manuscript on the 19th of december 2023. 

Comment 12

Overall comment on the Manuscript: From my point of view, the report lacks key data of the clinical case, mainly TFIp, reason for not performing surgery in the first place, reason for delivering 8 cycles of Carboplatin-Paclitaxel, maintenance treatment, and so on (tumoral markers at diagnosis, at recurrence diagnosis, etc). These pieces of information are key to understand or justify why a second cytoreductive surgery was proposed.

Nowadays, for a patient not included in a clinical trial, there is no justification to a) receive a second cytoreductive surgery aiming at the pleural and abdominal cavity and b) to receive HIPEC (nor HITOC), as recent clinical trials have failed to demonstrate a benefit in this context (Zivanovic, 2021), being only acceptable if performed in a “postCT” setting (CHIPOR Trial, Classe 2023). The authors did not justify correctly the rationale behind delaying 1mo the abdominal surgery after the thoracic surgery.

Response 12: The patient was experiencing respiratory failure with complete right lung collapse, making the HITHOC procedure the primary option.The one-month interval was arbitrary and was crucial for the patient's recovery following the "hitoc" procedure. Would have been the patient nowadays still alive? Certainly not, considering the patient's history of oncological treatment. The combined surgery played a vital role in her survival. Indeed, the patient would not have survived without the combined surgery, given her post-oncological treatment status. It is up to decide whether there is a benefit in survival without recurrence beyond a year compared to imminent death. The distinction between studies and case reports lies in the personalization of the patient's experience. In this case, the patient, a 58-year-old female patient, has a name, and these surgeries have proven beneficial for her and her family.

We used the MDPI English Editing Service, so no other issues regarding the english language should arrise.

Reviewer 2 Report

Comments and Suggestions for Authors

Moldovan and coworkers presented an interesting case report on the use of HITOC and HIPEC as a strategy against recurrent metastatic ovarian carcinoma. As a case report, the authors showed the successful results obtained in a single patient. Some points are suggested below to improve the quality of the paper:

A) It could be presented how much the patient is above the average of the current survival rate

B) Indicate in each figure, using arrows or letters, what is described in the legend. For example, peritoneal carcinomatosis (PC).

C) The images could be presented in a larger size on the pages.

D) Figure 8: Please include the scale bar.

E) Cite the supplementary material at some point in the text

Author Response

Comment 1: A) It could be presented how much the patient is above the average of the current survival rate

Response 1: We, accordingly, revised this matter. In our service, a significant milestone was reached, with a 40% 5-year survival rate for ovarian peritoneal carcinomatosis. The ongoing efforts in managing cases are highlighted by the current patient, who has had a 1.8-year survival without signs of recurrence, reflecting our commitment to advancing outcomes in ovarian peritoneal carcinomatosis.

Comment 2: B) Indicate in each figure, using arrows or letters, what is described in the legend. For example, peritoneal carcinomatosis (PC).

Response 2: We, accordingly, revised this matter.

Comment 3: C) The images could be presented in a larger size on the pages.

Response 3: We, accordingly, revised this matter.

Comment 4: D) Figure 8: Please include the scale bar.

Response 4: We, accordingly, revised this matter, as a suggestion of another reviewer and eliminated figure 8.

Comment 5: E) Cite the supplementary material at some point in the text

Response 5: We, accordingly, revised this matter.

Reviewer 3 Report

Comments and Suggestions for Authors

Congratulations to the authors on successful application of sequenced HIPEC and HITOC therapy and great result.

In the paper, there should be a brief introduction about HITOC and HIPEC before the case presentation. Also, I would suggest that in the introductory part of the case presentation, they should specify the type of primary ovarian tumor being discussed. There are no clear boundaries between the case presentation and the discussion, nor a distinct conclusion. It would be advisable to separate these sections to enhance the effectiveness of the written content.

I suggest creating a collage from the microphotographs in Figures 8 and 9, or omitting the images from Figure 8 as they are not of good quality (not sufficiently clear). Thw order of the appearance of Figures 8 and 9 an the text before Figures 6 and 7 should be corrected.

However, it would be beneficial to include several more recent references regarding the application of HITOC and HIPEC therapy.

Author Response

Comment 1: In the paper, there should be a brief introduction about HITOC and HIPEC before the case presentation. Also, I would suggest that in the introductory part of the case presentation, they should specify the type of primary ovarian tumor being discussed. There are no clear boundaries between the case presentation and the discussion, nor a distinct conclusion. It would be advisable to separate these sections to enhance the effectiveness of the written content.

Response 1: We, accordingly, addressed this matter. We added an introduction to our paper and a discussion and conclusion section which are now distinct to identify. We also detailed the case and added informations about the patients history with the disease and the complete course of treatment which she underwent.

Comment 2: I suggest creating a collage from the microphotographs in Figures 8 and 9, or omitting the images from Figure 8 as they are not of good quality (not sufficiently clear). Thw order of the appearance of Figures 8 and 9 an the text before Figures 6 and 7 should be corrected.

Response 2: We, accordingly, revised this matter. Figure 8 was eliminated and the text before figures 6 and 7 was corrected.

Comment 3: However, it would be beneficial to include several more recent references regarding the application of HITOC and HIPEC therapy.

Response 3: We, accordingly, revised this matter.

Round 2

Reviewer 3 Report

Comments and Suggestions for Authors

Congratulations to the authors on presenting this interesting case and on the successful application of sequenced HIPEC and HITOC therapy, as well as the outstanding results in treating advanced ovarian cancer. The authors have addressed all of my comments, and I believe that the paper, in its current form, can be accepted for publication.

Author Response

Question 1: Congratulations to the authors on presenting this interesting case and on the successful application of sequenced HIPEC and HITOC therapy, as well as the outstanding results in treating advanced ovarian cancer. The authors have addressed all of my comments, and I believe that the paper, in its current form, can be accepted for publication.

Response 1: Thank you for all support and guidance.